# Unlocking New Horizons in Small-Cell Lung Cancer Treatment: The Onset of Antibody–Drug Conjugates

**DOI:** 10.3390/cancers15225368

**Published:** 2023-11-10

**Authors:** Lorenzo Belluomini, Marco Sposito, Alice Avancini, Jessica Insolda, Michele Milella, Antonio Rossi, Sara Pilotto

**Affiliations:** 1Section of Innovation Biomedicine—Oncology Area, Department of Engineering for Innovation Medicine (DIMI), University of Verona and University and Hospital Trust (AOUI) of Verona, 37134 Verona, Italy; lorenzo.belluomini@univr.it (L.B.); marcosposito91@gmail.com (M.S.); alice.avancini@univr.it (A.A.); jessica.insolda@univr.it (J.I.); michele.milella@univr.it (M.M.); sara.pilotto@univr.it (S.P.); 2Therapeutic Science & Strategy Unit, Oncology Centre of Excellence, IQVIA, 20019 Milan, Italy

**Keywords:** SCLC, ADCs, TROP2, CEACAM5, DLL3

## Abstract

**Simple Summary:**

Small-cell lung cancer (SCLC) represents a highly aggressive form of lung cancer. Although SCLC initially responds to chemoimmunotherapy, it inevitably relapses. Antibody–drug conjugates (ADCs) have emerged as a promising treatment approach. ADCs consist of antibodies targeting specific tumor antigens, delivering cytotoxic drugs directly to cancer cells, and reducing side effects. In this review, we collected the available data regarding ADCs evaluated in SCLC. While some ADCs yielded negative results, others, such as those targeting B7-H3, SEZ6, CEACAM5, and TROP2 ADC, show promise. Further research is needed to determine ADCs’ role in SCLC treatment.

**Abstract:**

Small-cell lung cancer (SCLC) is a highly aggressive disease, accounting for about 15% of all lung cancer cases. Despite initial responses to chemoimmunotherapy, SCLC recurs and becomes resistant to treatment. Recently, antibody–drug conjugates (ADCs) have emerged as a promising therapeutic option for SCLC. ADCs consist of an antibody that specifically targets a tumor antigen linked to a cytotoxic drug. The antibody delivers the drug directly to the cancer cells, minimizing off-target toxicity and improving the therapeutic index. Several ADCs targeting different tumor antigens are currently being evaluated in clinical trials for SCLC. Despite the negative results of rovalpituzumab tesirine (Rova-T), other ADCs targeting different antigens, such as B7-H3, seizure-related homolog 6 (SEZ6), and CEACAM5, have also been investigated in clinical trials, including for SCLC, and their results suggest preliminary activity, either alone or in combination with other therapies. More recently, sacituzumab govitecan, an anti-TROP2 ADC, demonstrated promising activity in lung cancer, including SCLC. Furthermore, an anti-B7-H3 (CD276), ifinatamab deruxtecan (DS7300A), showed a high response rate and durable responses in heavily pretreated SCLC. Overall, ADCs represent an intriguing approach to treating SCLC, particularly in the relapsed or refractory setting. Further studies are needed to determine their efficacy and safety and the best location in the treatment algorithm for SCLC. In this review, we aim to collect and describe the results regarding the past, the present, and the future of ADCs in SCLC.

## 1. Introduction

Small-cell lung cancer (SCLC) is a highly aggressive neuroendocrine carcinoma, constituting around 13–15% of lung cancer cases. Patients often face an unfavorable prognosis, with a low 5-year survival rate [1].

Current treatments include platinum-based chemotherapy with radiation for limited-stage cases (LS SCLC) [2] and chemotherapy with a PD-L1 inhibitor for extensive-stage cases (ES SCLC) [3,4]. Long-term follow-up data suggest that extended treatment with an immune checkpoint inhibitor (ICI) and chemotherapy significantly improves survival, with three times more patients alive at 3 years compared to chemotherapy alone [4]. Thus, immunotherapy has shown promising results, especially in specific subgroups of patients, but unfortunately, disease progression inevitably occurs [3].

In this light, evaluating novel therapeutic possibilities that are effective in treating this prognostically bleak disease represents an urgent medical need. Antibody–drug conjugates (ADCs) are currently on the rise in various other histologies. Building upon their effectiveness, novel studies are assessing their potential therapeutic applications in the context of advanced SCLC treatment.

An ADC consists of a monoclonal antibody, a linker, and a payload. Initially, mouse-derived antibodies were utilized, leading to high failure rates due to severe immunogenicity-related side effects. With the advent of recombinant technology, murine antibodies were largely replaced by humanized antibodies. The primary component of antibodies employed in current ADCs is immunoglobulin G (IgG) [5]. Because of its ability to induce strong immune-related functions, IgG1 is the most frequently used immunoglobulin in ADCs [6]. The selection of a target antigen on tumor cells, the so-called tumor-associated antigen (TAA), serves as the guiding factor for ADC drugs to identify and enter tumor cells, thereby determining their therapeutic effects. To minimize off-target toxicity, the targeted antigen should preferably be predominantly expressed on the tumor cell surface [7].

The linker in ADCs connects the antibody with the cytotoxic drug. It plays a critical role in the stability of ADCs and the release profiles of payloads, consequently influencing the therapeutic index of ADCs. An ideal linker should limit premature payload release in plasma while facilitating the active release of drugs at the desired target sites. Most ADCs employ two types of linkers: cleavable and non-cleavable. The first ones are designed to cleave under specific conditions, such as in a low-pH environment or in the presence of proteolytic enzymes [5].

The cytotoxic payload is the active component that exerts cytotoxicity upon the internalization of ADCs into cancer cells. These compounds must remain stable under physiological conditions and possess functional groups suitable for conjugation with the antibody. To date, the cytotoxic payloads utilized in ADCs primarily comprise potent tubulin inhibitors and DNA-damaging agents [8,9]. In this light, the drug–antibody ratio (DAR) represents the average number of payload molecules (ranging from 2 to 8) attached to each monoclonal antibody. While ADCs with higher DARs exhibited greater potency in vitro, preclinical evidence demonstrated that they may have higher toxicity and a lower therapeutic index [10].

Following intravenous injection, ADCs bind to their target antigens and undergo receptor-mediated endocytosis for internalization within the cell. Once inside the cell, the cytotoxic payload carries out its cytotoxic role through its interaction with microtubules or DNA. In addition, apoptotic cells or ADCs may release the cytotoxic component into the extracellular space, potentially inducing cell death in neighboring cells, a phenomenon known as the “bystander effect” [11].

The current review aims to explore the past, the present, and the future of ADCs as therapeutic agents for the treatment of advanced SCLC (Figure 1).

## 2. Trophoblast Cell Surface Antigen (TROP2)

Trophoblast cell surface antigen (TROP2) is a transmembrane glycoprotein encoded by the tumor-associated calcium signal transducer 2 gene (TACSTD2) [12]. TROP2 is composed of an extracellular domain with a thyroglobulin type-1 repeat region and an epidermal growth factor-like region, as well as transmembrane and cytoplasmic domains [13]. The extracellular domain of TROP-2 can potentially bind to growth factors, such as insulin-like growth factor 1 (IGF-1) and other proteins involved in cellular processes [14]. TROP2 plays a role in several signaling pathways associated with cancer development, including calcium signaling, β-catenin signaling, and fibronectin adhesion [15,16,17]. In this light, the overexpression of TROP2 is linked to cancer cell growth and metastasis across various tumor types, including SCLC. On the other hand, no supporting evidence indicates that molecular alterations or mutations in TROP2 may trigger oncogenic transformation, suggesting that TROP2-expressing cancers could belong to the non-oncogene-addicted subtype. Interestingly, elevated TROP2 expression is present in 64% of lung adenocarcinomas, 75% of squamous cell lung cancers, and 18% of high-grade neuroendocrine lung carcinomas (including 10% of SCLC) [18]. Based on the current evidence, the role of increased TROP-2 expression as a predictive factor for the response to ADCs is uncertain. Cardillo et al. [19] demonstrated a positive correlation between high TROP-2 expression and treatment response in triple-negative breast cancer patients. However, this observation was not confirmed within the IMMU-132-01 trial, which was conducted in patients with advanced and pretreated non-small-cell lung cancer (NSCLC) [20].

### Anti-TROP2 ADCs

Sacituzumab govitecan (SG) is an ADC engineered to specifically target the TROP2 antigen. This ADC is composed of a humanized monoclonal antibody (hRS7 IgG1κ), which exhibits high specificity for recognizing TROP2. It is linked to a topoisomerase I inhibitor, SN-38, the active metabolite of irinotecan, through a hydrolyzable linker. Upon binding to the TROP2 receptor present on the surface of cancer cells, SG undergoes internalization into the cell. This internalization process triggers the hydrolysis of the linker, leading to the release of SN-38 within the cancer cell. Once released, SN-38 exerts its cytotoxic effects by interacting with topoisomerase I, an enzyme involved in the re-ligation of single-strand breaks generated by topoisomerase I during its normal cellular processes. This interference disrupts the DNA replication and repair mechanisms, ultimately inducing cellular apoptosis or programmed cell death [11] (Figure 2).

In a phase I/II study involving non-TROP2-selected patients with advanced epithelial cancers, SG showed promising antitumor effects and acceptable safety. In 62 patients enrolled affected with advanced SCLC, SG demonstrated an Objective Response Rate (ORR) of 17.7 (Confidence Interval (CI) 9.2–29.5), a median duration of response (DOR) of 5.7 months (CI 3.6–19.9), and a progression-free survival (PFS) and overall survival (OS) of 3.7 (CI 2.1–4.8) and 7.1 (CI 5.6–8.1) months, respectively [21]. Among all of the adverse events reported, the most commonly occurring were diarrhea, fatigue, neutropenia, anemia, and alopecia. Notably, grade 3–4 neutropenia was the primary reason for dose reductions [21,22] (Table 1).

The phase II study TROPiCS-03 (NCT03964727) is currently active and will evaluate SG’s safety and efficacy in patients with advanced solid tumors, including SCLC, pretreated with first-line therapy (Table 2).

## 3. Delta-like Ligand 3 (DLL3)

The Delta-like Ligand 3 (DLL3) gene encodes a protein that inhibits the Notch signaling pathway in SCLC cells. Notch signaling is involved in various cancer-related processes, including cell proliferation, differentiation, immune modulation, and resistance to chemotherapy [30]. In this light, the increased expression of DLL3 promotes SCLC cells’ growth, migration, and invasiveness. Moreover, DLL3 overexpression is linked to the development of metastatic and treatment-resistant characteristics in neuroendocrine cancers by promoting cell growth and resistance to chemotherapy [31].

In healthy subjects, low levels of DLL3 are present in the Golgi apparatus and cytoplasmic vesicles of normal cells [32]. However, up to 85% of SCLC exhibits the DLL3 protein at the membrane level. This abnormal expression is due to mechanisms not completely elucidated. Interestingly, a few studies have reported the preservation of DLL3 expression levels during chemotherapy, whereas others have observed dynamic fluctuations after treatments [33,34].

The differences in the expression patterns and subcellular distributions of DLL3 between normal and malignant cells underscore its considerable appeal as a therapeutic target characterized by an intrinsic propensity for tumor selectivity. In this light, several therapeutic strategies, such as Chimeric Antigen Receptor T-cell therapies (CAR-T cells) and bispecific T-cell engager (BiTE), are currently under investigation, whereas others, e.g., ADCs, have not demonstrated significant benefits in clinical trials.

### Anti-DLL3 ADCs

Rovalpituzumab tesirine (Rova-T) is a DLL3-specific humanized monoclonal antibody against SC16 linked to a membrane-permeable pyrrolobenzodiazepine (PBD) dimer toxin through a linker. After the binding of Rova-T to cell-surface DLL3, the ADC–target complex is internalized via endocytosis. Subsequently, the linker is cleaved by lysosome-associated cathepsin B, releasing PBD into the cytoplasm and into the nucleus, where PBD forms cross-links with DNA, ultimately triggering tumor cell apoptosis [35] (Figure 2). The first-in-human trial evaluating Rova-T in pretreated patients with SCLC demonstrated an encouraging 1-year survival rate (36%) [35] compared with previous results [36].

The TRINITY trial was an open-label, single-arm, phase II study that assessed the safety and efficacy of Rova-T in pretreated (three or more lines) patients with DLL3-expressing SCLC [23]. The ORR, one of the primary endpoints, was 12.4% for all patients, 14.3% for those classified as DLL3-high, and 13.2% for DLL3-positive patients. The mOS reached 5.6 months among all patients and extended to 5.7 months for those categorized as DLL3-high. The predominant AEs included fatigue, photosensitivity reactions, and pleural effusion. Notably, grade 3–5 AEs were observed in 213 patients, constituting 63% of the cohort [23].

In line with the modest benefit and the high toxicity rate demonstrated in the TRINITY trial, two other studies involving Rova-T showed no encouraging results. The TAHOE trial, a randomized phase 3 study comparing Rova-T with topotecan as second-line therapy in DLL3-high metastatic SCLC, did not reach the primary endpoint, demonstrating a median OS of 6.3 months (5.6–7.3) in the Rova-T arm and 8.6 months (7.7–10.1) in the topotecan arm (Hazard Ratio (HR) = 1.46, [CI 1.17–1.82]) [24]. Similarly to the TRINITY trial, it is noteworthy to highlight the significant findings in the Rova-T group, with a 42% incidence of grades 3–4 and a 22% incidence of grade 5 treatment adverse events (TAEs). Based on these negative results, enrollment was discontinued.

Another phase 3 study (MERU trial) was terminated early due to the lack of survival benefits and a high rate of adverse events (grade ≥ 3) [25]. In the study, patients without disease progression after four cycles of platinum-based front-line chemotherapy were randomized in a 1:1 ratio to receive Rova-T or a placebo. Unfortunately, the study failed to achieve its primary endpoint due to the absence of a survival advantage in the Rova-T treatment group. The peculiar toxicity of Rova-T (or similar ADCs with PBD payload) is primarily due to two potential mechanisms: the premature cleavage of the ADC linker and a “bystander” effect, where the cytotoxic component affects healthy cells when it diffuses from dying tumor cells [37].

A phase 1 study evaluated the safety, tolerability, pharmacokinetics, and preliminary efficacy of SC-002, an ADC directed against DLL3 and conjugated with PBD, in patients with advanced SCLC and large-cell neuroendocrine carcinoma. Among the 35 enrolled patients who received at least one dose of SC-002, 66% experienced serious adverse events, with 37% considered related to SC-002. Grade 3 and 4 adverse events were observed in 60% and 6% of patients, with effusion and hypoalbuminemia being the most common. One patient experienced a grade 5 adverse event. Partial responses were achieved in 14% of patients, while no complete responses were observed [26]. In line with the results obtained with Rova-T, SC-002 demonstrated limited activity and high toxicity in pretreated SCLC (Table 1).

## 4. B7-H3 (CD276)

B7-H3, also known as CD276, is a member of the B7 superfamily of immune checkpoint molecules. B7-H3 is a transmembrane protein with an extracellular domain composed of two identical domains (4IgB7-H3 isoforms) and a short intracellular tail [38]. This multifaceted molecule plays a central role in shaping the landscape of T-cell-mediated immune responses. Its intricate functions encompass both co-stimulatory and co-inhibitory roles, making it a crucial regulator in the immune reactions mediated by T cells [38].

B7-H3 is overexpressed in a broad spectrum of cancer histologies. This overexpression appears to be associated with advanced tumor stages and higher tumor grades in various cancers, such as endometrial, cervical, breast, and kidney cancer and oral squamous cell carcinoma. In this light, this elevated expression of the B7-H3 protein has been linked to increased cancer cell proliferation and enhanced invasive potential in several cancer types, including pancreatic, breast, colorectal, and prostate cancer.

Interestingly, the role of B7-H3 in predicting patient outcomes is still not clear. In many cancer types, its overexpression has been consistently associated with a less favorable prognosis [39,40,41]. However, in gastric and pancreatic cancer, high B7-H3 expression has been linked to improved patient survival [42,43]. To understand these disparities, several factors must be considered. The differences in cancer types or subtypes, the complexity of tumor heterogeneity, variations in sample sizes, variations in clinical stages, the timing of B7-H3 measurement during the course of the disease (and treatment), and the diverse methodologies employed in different research studies all contribute to the intricate puzzle of B7-H3’s role in cancer.

### Anti-B7-H3 ADCs

Ifinatamab deruxtecan (I-DXd, DS-7300) is a targeted ADC against B7-H3, composed of humanized anti–B7-H3 IgG1 conjugated to deruxtecan (Dxd), a topoisomerase I inhibitor, through a cleavable tetrapeptide linker. Upon binding to B7-H3, the ifinatamab deruxtecan complex undergoes internalization and linker cleavage within the intracellular environment by lysosomal enzymes that are overexpressed in tumor cells. Following its release, DXd, which is permeable to cell membranes, induces DNA damage and cell death through apoptosis (Figure 2). DXd, a derivative of exatecan, is approximately 10 times more potent than SN-38.

At present, two studies are currently investigating I-Dxd in SCLC. A phase 1/2 study (NCT04145622) is being conducted in the United States and Japan and focuses on I-Dxd in patients with specific advanced solid tumors (including SCLC) [44]. The study comprises two phases: dose escalation (part 1) and dose expansion (part 2). The primary goals are to assess the safety, tolerability, and effectiveness of I-Dxd and identify the maximum tolerated or recommended dose for the expansion phase.

As of 31 January 2023, of the 21 SCLC patients, 11 demonstrated an ORR, accounting for 52% of the cohort, comprising 1 complete response (CR) and 10 partial responses (PRs). The median duration of response (DOR) was 5.9 months (95% CI 2.8–7.5). Median PFS was 5.8 months (95% CI, 3.9–8.1), while median OS reached 9.9 months (95% CI, 5.8–not reached) [27].

At the June 30, 2022, data cutoff, the 16.0 mg/kg cohort was closed for safety reasons, with high rates of serious and grade ≥3 treatment-emergent adverse events (TEAs). Anemia (19%), neutropenia (4%), nausea (3%), pneumonia (3%), and decreased neutrophil count (3%) were the most common (23%) grade ≥3 TEAs. ILD/pneumonitis was present in nine patients, of which seven (grade 1, *n* = 2; grade 2, *n* = 4; grade 5, *n* = 1) were adjudicated as drug-related ILD [27] (Table 1).

A phase 2, randomized, open-label study (NCT05280470) is currently evaluating the efficacy, safety, and pharmacokinetics of I-Dxd in patients with pretreated (at least one prior line of platinum-based chemotherapy and a maximum of three prior lines of therapy) metastatic SCLC (Table 2).

## 5. Seizure-Related Homolog 6 (SEZ6)

Seizure-related homolog 6 (SEZ6) is a transmembrane protein localized to the cell surface of specific neuronal lineage cells. In murine models, the SEZ6 gene family demonstrated a regulatory influence over various aspects of neural function. Specifically, this gene family has been observed to modulate synaptic density, synaptic plasticity, and dendritic morphology within the cortical and hippocampal regions. Furthermore, SEZ6 seems to influence neuronal connectivity in the cerebellum [45].

Interestingly, SEZ6 exhibits prominent expression within the majority of SCLC cases, as well as in other neuroendocrine and central nervous system (CNS) tumors. Conversely, SEZ6 demonstrates low expression in normal tissues and non-neuroendocrine tumor types. This expression pattern underlines SEZ6 as an attractive therapeutic target, offering the potential to maximize treatment efficacy while minimizing toxicity [46].

### Anti-SEZ6 ADCs

ABBV-011 is an anti-SEZ6 monoclonal antibody conjugated via a non-cleavable linker drug (LD19.10) to calicheamicin, a cytotoxic payload that induces double-strand breaks in the DNA (Figure 2). Interestingly, calicheamicin has been previously employed in two other ADC therapies, gemtuzumab ozogamicin and inotuzumab ozogamicin (both approved by the FDA for hematological malignancies) [47].

A first-in-human, phase 1 study (NCT03639194) is currently evaluating the safety and tolerability of ABBV-011 as a single agent or in combination with budigalimab (ABBV-181, an anti-PD1 monoclonal antibody) in participants with relapsed or refractory SCLC. As of the data cutoff date, August 2022, a total of 99 patients had undergone treatment with ABBV-011 monotherapy. Considering a median duration of treatment of 12 weeks (from 1.9 to 63.3 weeks), the ORR was 25%, with 10 partial responses (PRs). The median duration of response was 4.2 months (CI 95%, 2.6–6.7), and the median PFS was 3.5 months. The clinical benefit rate (CBR) reached 65%, with 10 PRs and 16 cases of stable disease. Notably, a CBR lasting beyond 12 weeks was observed in 43% of cases. The most prevalent treatment-emergent adverse events (TEAEs) included fatigue (48%), nausea (45%), anorexia (38%), thrombocytopenia (38%), and vomiting (35%). Grade 3 TEAEs were documented in 18 patients (45%), with the most common occurrences being fatigue, thrombocytopenia, and neutropenia (each at 10%) [28] (Table 1).

In addition, a novel anti-SEZ6 ADC, ABBV-706, is currently under investigation in a phase 1 clinical trial (NCT05599984). This trial aims to evaluate the efficacy and safety of ABBV-706, both as monotherapy and in combination with budigalimab, cisplatin, or carboplatin in patients with advanced solid tumors (including SCLC) (Table 2).

## 6. Carcinoembryonic Antigen-Related Cell Adhesion Molecule 5 (CEACAM5 or CD66e)

Carcinoembryonic antigen-related cell adhesion molecule 5 (CEACAM5), alternatively known as CD66e (Cluster of Differentiation 66e), belongs to the carcinoembryonic antigen (CEA) gene family, and it is involved in cell adhesion, signaling, and the promotion of cancer progression and metastasis [48,49].

CEACAM5 is overexpressed in numerous epithelial tumors, especially in lung cancers [50]. However, despite exhibiting distinct expression patterns in various tissues (low expression levels in normal tissue), CEACAM5 shares structural homology and sequence similarity with other CEACAMs. Consequently, any molecules that target CEACAM5 must exhibit a high degree of binding specificity to individual CEACAMs to prevent cross-reactivity and the associated dose-limiting toxicity [51].

### Anti-CEACAM5 ADCs

Tusamitamab ravtansine (SAR408701) is a targeted ADC against tumor cells expressing CEACAM5, composed of a humanized anti-CEACAM5 monoclonal antibody covalently linked to the potent cytotoxic agent, maytansinoid DM4, via a cleavable linker [50]. The antibody component of SAR408701 binds to the extracellular domain of CEACAM5, initiating the internalization of the ADC into the tumor cell, followed by the subsequent release of DM4 within the cytoplasm. DM4 elicits its cytotoxic effects by disrupting microtubule assembly, inducing cell cycle arrest, and initiating apoptosis. Notably, both DM4 and its biologically active derivative, S-methyl-DM4, are capable of traversing cell membranes. Consequently, the cytotoxicity against tumor cells is twofold, both directed toward CEACAM5-expressing tumor cells and enhanced through the bystander effect (the potential release of the cytotoxic component into the extracellular space, with subsequent bystander cell death) [50] (Figure 2).

A first-in-human, phase 1 study (NCT02187848) is currently evaluating the safety and tolerability of SAR408701 in patients with advanced solid tumors, including, to date, only one patient affected by SCLC. Twenty-two patients (71%) experienced ≥1 TEAE, seven patients (22.6%) experienced ≥1 treatment-related grade ≥3 TEAE, and three patients (9.7%) discontinued treatment due to TEAEs (Table 1). The most common TEAEs were asthenia (25.8%), decreased appetite (25.8%), keratopathy (25.8%), and nausea (25.8%). Three patients (9.7%) had objective responses, eleven patients (35.5%) had stable disease, and thirteen patients (41.9%) had progressive disease [52]. Considering the evaluation of safety characteristics and pharmacokinetic information, the maximum tolerated dose (MTD) for tusamitamab ravtansine was established as 100 mg/m^2^ administered every 2 weeks (Q2W). In the same trial, two distinct treatment schedules were also investigated, and SAR408701 exhibited a favorable safety profile with both of these alternative dosing regimens. The MTDs were identified as 170 mg/m^2^ (Loading Dose), followed by 100 mg/m^2^ administered every 2 weeks (Q2W), and 170 mg/m^2^ administered every 3 weeks (Q3W) as a fixed dose [29]. 

## 7. Future Perspectives

The development of antibody–drug conjugates (ADCs) is continuously progressing, and new potential therapies are constantly evolving, leveraging novel payloads/targets/linkers.

A phase 1/2a study (NCT03221400) is currently evaluating PEN-866 in solid cancers, including SCLC. PEN-866 is an ADC that specifically targets intracellular Heat Shock Protein 90 (HSP90) and carries the topoisomerase 1 inhibitor SN-38 as a payload. PEN-866 demonstrated preliminary evidence of antitumor activity in preclinical models and in a phase I clinical trial [53,54].

Recently, Yamaguchi et al. demonstrated that junctional adhesion molecule 3 (JAM3) is expressed in SCLC cell lines and tissues [55]. Moreover, the use of an anti-JAM3 monoclonal antibody, HSL156, associated with a recombinant protein called DT3C (derived from the diphtheria toxin) hindered the growth of all examined SCLC cell lines, suggesting a novel (potential) ADC approach [55].

Another potential ADC target is the cell adhesion molecule neurexin-1 (NRXN1). NRXN1 is a transmembrane protein specifically overexpressed in SCLC and shows minimal to no expression in normal tissues. Using a primary anti-NRXN1 monoclonal antibody in combination with a secondary ADC demonstrated antitumor activity in SCLC cell lines [56].

Recently, Kim et al. developed an ADC (4C9-DM1) targeting c-KIT with a microtubule inhibitor payload, which showed antitumor efficacy in SCLC cell lines and mouse xenografts [57].

## 8. Discussion

Despite the recent improvements in the development of SCLC treatments, this histological subtype continues to exhibit a poor prognosis. The introduction of immunotherapy in combination with chemotherapy has significantly improved the survival rates of patients with SCLC [3,4]. On the other hand, the absence of reliable biomarkers that may predict a positive response to molecular-targeted therapies poses a significant challenge. In this light, further research is necessary to achieve additional survival benefits, particularly after the failure of first-line chemoimmunotherapy. Among the latest developments, ADCs may seem to offer a promising opportunity.

ADCs, which combine the precision of monoclonal antibodies with the cytotoxic potential of chemotherapeutic agents, have already exhibited noteworthy success in treating several malignancies, including breast cancer [58,59], bladder cancer [60], and lung cancer [61].

Only preliminary data are available regarding the role of ADCs in SCLC. Alongside trials with moderately promising results, such as those concerning SG [21], some studies showed less favorable outcomes, e.g., Rova-T, which exhibited modest efficacy and demonstrated significant and potentially life-threatening toxicities [23,24].

Patient selection is an important aspect worth exploring, particularly in identifying predictive factors that could anticipate treatment response. Despite the absence of available molecularly targeted therapies in SCLC, the application of ADCs may represent a potential treatment strategy that takes advantage of the intrinsic molecular characteristics exhibited by the tumor cells, such as the overexpression of TROP2, DLL3, B7-H3, and CECAM5.

However, to date, the studies available have provided conflicting data regarding the overexpression of specific markers as predictors of treatment response. In this light, the overexpression of TROP2 has not been definitively correlated with improved therapeutic responses in patients with advanced NSCLC who have undergone anti-TROP2 ADC treatment [20]. On the other hand, a positive correlation between high TROP-2 expression and treatment response in advanced triple-negative breast cancer has been demonstrated [19]. In addition, initially, patients with high DLL3 expression (≥50% of tumor cells) showed better response rates and disease control than those with low DLL3 expression [35]. However, follow-up studies, including those targeting exclusively high-DLL3 patients, did not confirm the overexpression of DLL3 as a predictive biomarker [24].

ADCs are designed to selectively deliver cytotoxic payloads specifically within the confines of neoplastic cells while sparing healthy neighboring tissues. However, studies conducted on breast cancer and NSCLC have brought to light substantial toxicities, ranging from ocular complications [62] to the development of pulmonary interstitial diseases [58,61]. Consequently, these adverse events pose a fresh set of challenges to clinicians, given the relative novelty of ADCs’ unique toxicity profiles. The intricate relationships between several factors may contribute to the development of adverse events associated with ADC-based therapies. Firstly, the affinity of the monoclonal antibody component of ADCs for the target antigen plays a pivotal role. The strength and specificity of this binding interaction determine the selectivity of the ADC for cancer cells. While a high affinity ensures the precise targeting of tumor cells, it also raises the likelihood of off-target effects, potentially affecting normal tissues expressing the same antigen [63]. Moreover, the linker molecule, which acts as the bridge connecting the antibody and the cytotoxic payload, also influences the overall toxicity profile of an ADC. The linker must be stable while circulating in the bloodstream, ensuring that the cytotoxic payload remains attached to the antibody until it reaches its target. However, it should also be capable of efficiently releasing the drug once the ADC is internalized by the cancer cell. Inadequate linker stability or inefficient drug release can result in suboptimal therapeutic outcomes or increased off-target toxicity. Lastly, the inherent cytotoxicity of the drug conjugated to the ADC is a crucial determinant of its safety profile. Their cytotoxic effects are not limited to cancer cells alone. The drug may also exert its toxic effects on adjacent healthy cells, potentially leading to side effects [63].

Overall, the novelty of ADCs means that their behavior, in terms of both efficacy and potential side effects, may not always conform to conventional expectations. Consequently, further investigations are unquestionably required to elucidate the mechanisms of action and toxicity of these specific drugs. In addition, it will be essential in the future to optimize the management of adverse events by developing ADCs with more favorable toxicity profiles and, on the other hand, by optimizing supportive therapy, potentially to prevent the most common AEs.

Of note, the development of these drugs is taking place within a broader landscape of significant advancements in oncological therapeutics. Specifically, there is a growing emphasis on harnessing immune-related mechanisms to devise more effective treatments for several histologies. In this light, in a preclinical SCLC model, the combination of Rova-T with an anti-PD-1 agent showed promising results by enhancing tumor control and the immune response [64]. Subsequently, a clinical trial combining Rova-T with Nivolumab and Ipilimumab in heavily treated patients with ES SCLC reported a 30% response rate [65]. However, this combination raised concerns due to high toxicity rates, with half of the high-dose patients experiencing severe side effects, probably Rova-T-related [65].

Within this rapidly evolving field, two notable and up-and-coming areas of research and development are represented by Chimeric Antigen Receptor T-cell (CAR-T cell) and bispecific T-cell engager (BiTE). CAR T-cell therapy has demonstrated remarkable success in hematological malignancies, leading to durable remissions [66,67]. Ongoing research is expanding the evaluation of CAR T-cell therapy to a broader range of solid tumors, including SCLC [68,69]. For instance, TROP2/PD-L1 CAR-T cells exhibited promising results in vitro when tested against gastric cancer cell lines [70].

On the other hand, bispecific antibodies have shown great promise in activating the immune response against cancer and are currently being explored in various cancer types. Of note, Tarlatamab, a bispecific T-cell engager molecule that binds to both DLL3 and CD3, initiating T-cell-mediated apoptosis, demonstrated encouraging results and manageable toxicity in ES SCLC [71].

These developments exemplify the growing understanding of the immune system’s role in cancer control and the innovative strategies devised to exploit this knowledge.

In this light, the proper role and placement of ADCs in SCLC need to be clarified, considering the forthcoming results of the ongoing trials and the emerging alternatives (e.g., CAR T-cell therapy, BiTE) in the treatment landscape.

## 9. Conclusions

SCLC represents one of the most lethal solid cancers, requiring effective and safe therapeutic development. Within this context, ADCs emerge as a potential therapeutic alternative. The evidence in this field is steadily accumulating, with the progressive expansion of the target molecules and cytotoxic agents employed. Further studies, together with the results of ongoing trials, are eagerly awaited to better evaluate the efficacy and safety of ADCs in order to reshape the current treatment landscape of SCLC.

## Figures and Tables

**Figure 1 cancers-15-05368-f001:**
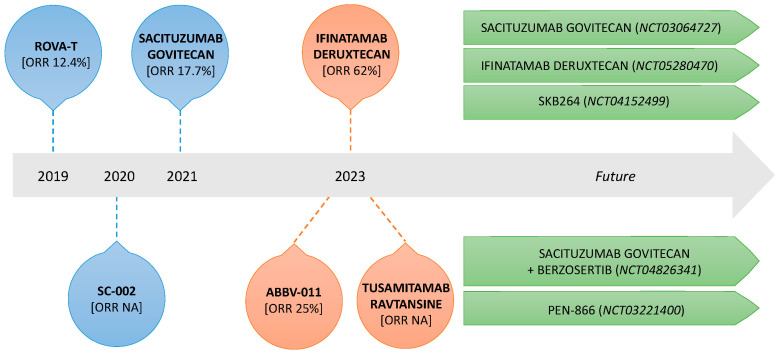
Timeline of key discoveries in ADC development (blue and orange circles) and ongoing trials in SCLC (green arrows).

**Figure 2 cancers-15-05368-f002:**
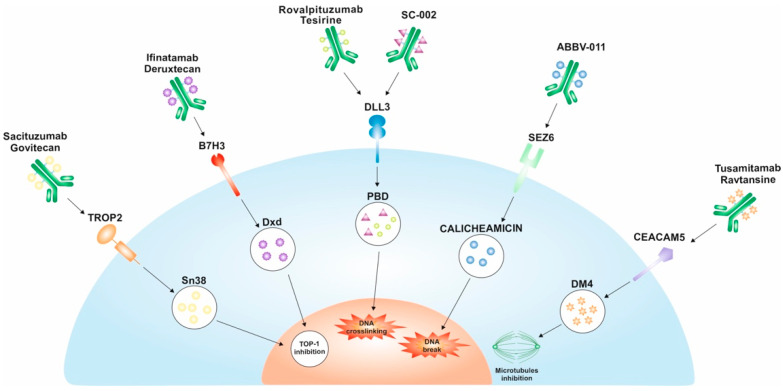
Mechanism of action of ADCs evaluated in SCLC.

**Table 1 cancers-15-05368-t001:** Efficacy and safety of the main ADCs in SCLC.

Trial	ADC	Target	Payload	SCLC (N)	Activity	Toxicity (≥G3)
Bardia et al. [21] IMMU-132-01	Sacituzumab govitecan	TROP2	SN–38	495	ORR 17.7% (CI 9.2–29.5) mDOR 5.7 mo (CI 3.6–19.9) mPFS 3.7 mo (CI 2.1–4.8) mOS 7.1 mo (CI 5.6–8.1)	Total 59.6%: - Neutropenia (42%) - Anemia (10%)
Morgensztern et al. [23] TRINITY	Rovalpituzumab tesirine	DLL3	Pyrrolobenzodiazepine	261	ORR 12.4% (CI 9.1–16.4) mOS 5.6 mo (CI 4.9–6.1) mPFS 3.5 mo (CI 3.0–3.9)	Total 40%: - Thrombocytopenia (11%) - Photosensitivity (7%) - Anemia (4%)
Blackhall et al. [24] TAHOE				296	mOS 6.3 mo (CI 5.6–7.3) mPFS 3 mo (CI 2.9–3.6) ORR 15%	Total 64%: - Thrombocytopenia (9%) - Dyspnea (7%) - Anemia (7%)
Johnson et al. [25] MERU				372	mOS 8.5 mo (CI 7.3–10.2) mPFS 3.7 mo (CI 2.9–4) ORR 9%	Total 59%: - Thrombocytopenia (9%) - Pleural effusion (4%) - Photosensitivity (4%)
Morgensztern et al. [26]	SC-002	DLL3	Pyrrolobenzodiazepine	35 ^§^	PR 14% CR 0% SD 40% PD 31%	Total 69%
Johnson et al. [27]	Ifinatamab deruxtecan	B7-H3	Deruxtecan	21	ORR 52% (CI 29.8–74.3%) mDOR 5.9 mo (CI 2.8–7.5) mPFS 5.8 mo (CI 3.9–8.1) mOS 9.9 mo (CI 5.8–NR)	Total 23% *: - Anemia (19%) - Neutropenia (4%) - Nausea (3%) - Pneumonia (3%)
Morgensztern et al. [28]	ABBV-011	SEZ6	Calicheamicin	99	ORR 25% mDOR 4.2 mo (CI 2.6–6.7) mPFS 3.5 mo CBR 65%	Total 45%: - Fatigue (10%) - Thrombocytopenia (10%) - Neutropenia (10%)
Tabernero et al. [29]	Tusamitamab ravtansine	CECAM5	Maytansinoid DM4	1	NA	Total 22.6%

Legend: N: number; mo: months; ORR: Objective Response Rate; mDOR: median duration of response; mPFS: median progression-free survival; PFS: progression-free survival; mOS: overall survival; OS: overall survival; PR: partial response; CR: complete response; SD: stable disease; PD: progressive disease; G: grade; CI: Confidence Interval; NA: Not Available. * Of the overall population, including SCLC; ^§^ including large-cell neuroendocrine lung carcinoma (LCNEC).

**Table 2 cancers-15-05368-t002:** Ongoing trials with ADCs in SCLC.

ClinicalTrial.gov ID	Phase	Setting	N	Treatment	Target	Primary Endpoint	Selected Secondary Endpoints	Status
NCT03964727	II	Pretreated solid tumors, including SCLC	165	Sacituzumab govitecan	Trop2	ORR	DOR, PFS, CBR, OS	Active, not recruiting
NCT05280470	II	Pretreated SCLC	91	Ifinatamab deruxtecan	B7-H3	ORR	TEAEs, PFS, OS, DOR	Active, not recruiting
NCT04152499	I–II	Pretreated solid tumors, including SCLC	430	SKB264	Trop2	MTD, RDE, ORR	DLTs, DOR, PFS, OS	Recruiting
NCT04826341	I–II	Pretreated solid tumors, including SCLC, small-cell neuroendocrine cancers, and HRD solid cancer	85	Sacituzumab govitecan + berzosertib	Trop2 + ATR inhibitor	MTD, ORR	PFS, OS, DOR	Recruiting
NCT03221400	I–IIa	Solid tumors	340	PEN-866	HSP90	DLT, ORR	DCR, PFS, OS, DOR	Recruiting

Legend: ES: Extended Stage; HRD: Homologous Recombination Deficient; SCLC: small-cell lung cancer; ORR: Objective Response Rate; CBR: clinical benefit rate; DOR: duration of response; MTD: maximum tolerated dose; RDE: recommended dose for expansion; PFS: progression-free survival; OS: overall survival; DCR: Disease Control Rate, DLT: dose-limiting toxicity; TEAEs: treatment-emergent adverse events.

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
