# Peer review of "Unlocking New Horizons in Small-Cell Lung Cancer Treatment: The Onset of Antibody–Drug Conjugates"

_cancers, 2023, doi:10.3390/cancers15225368_

Round 1
Reviewer 1 Report
Comments and Suggestions for Authors
In this review, the authors review past and ongoing clinical trials on ADC in SCLC, The review's thematic focus is interesting, as the SCLC issue is rarely addressed independently of NSCLC. After a brief presentation of the pathology and of ADC, the authors are presenting the tumor-associated antigen targeted by the current ADCs and the available associated clinical results.
The review is well-structured, rather well written and easy to read.
However to my opinion, the review does not totally meet the expectations from the title and the introduction (lanes 60-61), in particular concerning the future.
The authors remain on a very general level concerning the improvement of ADC (antibody, linker, payload), but what about current studies on improving this strategy in this disease? Are there any ongoing preclinical studies? New potential targets? Same target but different payload? If so, it could be of interest to add a paragraph on future directions through ongoing pre-clinical studies to “unlock new horizons”.
Despite subsequent advances in the design of ADC, toxicity remains a key issue; in this respect what should be, to the authors ‘opinion, the future directions in ADC in SCLC ?
For some targets, the authors mention other therapeutic strategies targeting the same TAA (DLL3), thus validating the target in SCLC or other cancers; what about the others, TROP-2?
What is the reason for the slow progress of targeted therapies in SCLC? The absence of reliable biomarkers allowing selection of eligible patients?
These points are more comments than request but answering them would enrich the review.
Minor corrections:
- A figure showing a timeline of key discovery in the ADC development in the field of SCLC in a figure would be relevant to be in line with “past, present and future”.
- The references in table 1 have to be corrected (23 instead of 24; ..)
- The legends of tables1 and 2 have to be corrected also: some terms are defined but are not in the table and vice versa.
- In table 1, is the term “safety” adequate to describe adverse events? Rather toxicity?
- In Table 2, a clinical trial is referenced (HSP90) without mention in the text
- Ifinatamab instead of infinatamab
Comments on the Quality of English LanguageThe english is readable with few minor spelling and syntax errors
Author Response
Dear Reviewer 1,
Please find enclosed the revised manuscript entitled “Unlocking new horizons in Small Cell Lung Cancer treatment: the onset of Antibody-Drug Conjugates”.
We would like to express our gratitude for your positive feedback and extend our thanks for your punctual comments, which have guided us in understanding the aspects that need improvement in our manuscript.
We marked the changes using a different color (yellow) of text (in the manuscript file). We also updated the tables and the figures.
All the revisions suggested were performed as a point-by-point rebuttal, as follows:
Reviewer #1: In this review, the authors review past and ongoing clinical trials on ADC in SCLC. The review's thematic focus is interesting, as the SCLC issue is rarely addressed independently of NSCLC. After a brief presentation of the pathology and of ADC, the authors are presenting the tumor-associated antigen targeted by the current ADCs and the available associated clinical results.
The review is well-structured, rather well written and easy to read.
Response: Thank you for appreciating our manuscript.
1 - The review does not totally meet the expectations from the title and the introduction (lanes 60-61), in particular concerning the future. The authors remain on a very general level concerning the improvement of ADC (antibody, linker, payload), but what about current studies on improving this strategy in this disease? Are there any ongoing preclinical studies? New potential targets? Same target but different payload? If so, it could be of interest to add a paragraph on future directions through ongoing pre-clinical studies to “unlock new horizons”.
Response: We added a paragraph (Future Perspective) on the main preclinical studies involving ADC in SCLC (lines 352-373).
2 - Despite subsequent advances in the design of ADC, toxicity remains a key issue; in this respect what should be, to the authors ‘opinion, the future directions in ADC in SCLC?
Response: We added a comment on that topic in the discussion paragraph (lines 431-434).
3 - For some targets, the authors mention other therapeutic strategies targeting the same TAA (DLL3), thus validating the target in SCLC or other cancers; what about the others, TROP-2?
Response: We added a detail on anti-TROP2 strategies in lines 449-450.
4 - What is the reason for the slow progress of targeted therapies in SCLC? The absence of reliable biomarkers allowing selection of eligible patients?
These points are more comments than request but answering them would enrich the review.
Response: Thank you for the suggestions. We added a comment on that topic in the discussion paragraph (lines 378-380).
5 - A figure showing a timeline of key discovery in the ADC development in the field of SCLC in a figure would be relevant to be in line with “past, present and future”.
Response: Thank you for the suggestion. We added a timeline of key discovery in the ADC development in SCLC as Figure 1.
6 - The references in table 1 have to be corrected (23 instead of 24; ..)
Response: We corrected the citations in table 1 (highlighted in yellow).
7 - The legends of tables1 and 2 have to be corrected also: some terms are defined but are not in the table and vice versa.
Response: We corrected the legends in table 1 and 2 (highlighted in yellow).
8 - In table 1, is the term “safety” adequate to describe adverse events? Rather toxicity?
Response: We changed the title to Toxicity.
9 - In Table 2, a clinical trial is referenced (HSP90) without mention in the text
Response: We added a comment on that trial in the future perspective paragraph (lines 356-360)
10 - Ifinatamab instead of infinatamab
Response: Thank you for the comment, we corrected the spelling.
Reviewer 2 Report
Comments and Suggestions for Authors
1.Sentence 55“"The mechanism of action involves the ADCs binding to the TAA, expressed on the 55 cell surface, and subsequently being internalized through endocytosis."” needs to be revised to make the sentence structure complete.
2.To enhance readability and clarity, it is crucial to make significant improvements in the language used throughout the article.
3.The inconsistent usage of "B7-H3" and "B7H3" throughout the article needs to be addressed and corrected for consistency.
4.Consistency is needed in placing the reference numbers, either within the sentence or at the end, to maintain a cohesive structure throughout the article.
Comments on the Quality of English LanguageI have noticed many inconsistencies in tenses, subject-verb agreement, singular/plural forms, and incomplete sentences in the provided text. Please carefully revise them throughout the text for better fluency and coherence.
Author Response
Dear Reviewer 2,
Please find enclosed the revised manuscript entitled “Unlocking new horizons in Small Cell Lung Cancer treatment: the onset of Antibody-Drug Conjugates”.
We would like to express our gratitude for your positive feedback and extend our thanks for your punctual comments, which have guided us in understanding the aspects that need improvement in our manuscript.
We marked the changes using a different color (yellow) of text (in the manuscript file). We also updated the tables and the figures.
All the revisions suggested were performed as a point-by-point rebuttal, as follows:
Reviewer #2:
1 - “"The mechanism of action involves the ADCs binding to the TAA, expressed on the cell surface, and subsequently being internalized through endocytosis."” needs to be revised to make the sentence structure complete.
Response: Thank you for the comment; we changed the entire paragraph.
2 - To enhance readability and clarity, it is crucial to make significant improvements in the language used throughout the article.
Response: We extensively revised and improved the English language along entire manuscript.
3 - The inconsistent usage of "B7-H3" and "B7H3" throughout the article needs to be addressed and corrected for consistency.
Response: Thank you for the comment; we uniformed with the use of “B7-H3” term throughout the text.
4 - Consistency is needed in placing the reference numbers, either within the sentence or at the end, to maintain a cohesive structure throughout the article.
Response: We corrected the references by placing them at the end of the sentences. Some were left within the sentence to better emphasize the reference to the source.
Reviewer 3 Report
Comments and Suggestions for Authors
This manuscript provides an insightful examination of the current state of antibody-drug conjugates (ADCs) in the context of Small Cell Lung Cancer (SCLC), with a focus on their evaluation in clinical trials. In comparison to traditional compounds, ADCs offer numerous advantages for cancer treatment, representing a rapidly evolving approach in the treatment of SCLC.
It's worth noting that other research groups have also explored the use of ADCs in lung cancer, including SCLC (DOI: 10.1200/EDBK_389968). To enhance the manuscript's contribution, it is essential to clearly articulate the novelty of the findings or perspective presented herein.
Additional suggestions for improving the manuscript include an expansion of the introduction section. This should encompass a more comprehensive overview of ADCs, delving into their structures, linkers, and the mechanisms by which ADCs operate in cancer therapy, among other relevant aspects. This would provide a stronger foundation for readers, enabling a deeper understanding of the topic.
Comments on the Quality of English LanguageNone
Author Response
Dear Reviewer 3,
Please find enclosed the revised manuscript entitled “Unlocking new horizons in Small Cell Lung Cancer treatment: the onset of Antibody-Drug Conjugates”.
We would like to express our gratitude for your positive feedback and extend our thanks for your punctual comments, which have guided us in understanding the aspects that need improvement in our manuscript.
We marked the changes using a different color (yellow) of text (in the manuscript file). We also updated the tables and the figures.
All the revisions suggested were performed as a point-by-point rebuttal, as follows:
Reviewer #3: This manuscript provides an insightful examination of the current state of antibody-drug conjugates (ADCs) in the context of Small Cell Lung Cancer (SCLC), with a focus on their evaluation in clinical trials. In comparison to traditional compounds, ADCs offer numerous advantages for cancer treatment, representing a rapidly evolving approach in the treatment of SCLC.
Response: Thank you for appreciating our manuscript.
1 - It's worth noting that other research groups have also explored the use of ADCs in lung cancer, including SCLC (DOI: 10.1200/EDBK_389968). To enhance the manuscript's contribution, it is essential to clearly articulate the novelty of the findings or perspective presented herein.
Response: We added the suggested reference, and we better articulated the novelty of the findings.
2 - Additional suggestions for improving the manuscript include an expansion of the introduction section. This should encompass a more comprehensive overview of ADCs, delving into their structures, linkers, and the mechanisms by which ADCs operate in cancer therapy, among other relevant aspects. This would provide a stronger foundation for readers, enabling a deeper understanding of the topic.
Response: Thank you for the comment; we added an in-depth description of ADC structures and mechanisms in the Introduction paragraph.